# Computed and Subjective Blue Scleral Color Analysis as a Diagnostic Tool for Iron Deficiency: A Pilot Study

**DOI:** 10.3390/jcm8111876

**Published:** 2019-11-05

**Authors:** Hervé Lobbes, Julien Dehos, Bruno Pereira, Guillaume Le Guenno, Laurent Sarry, Marc Ruivard

**Affiliations:** 1Internal Medicine Department, University Hospital Clermont-Ferrand, 1 place Lucie et Raymond Aubrac, 63003 Clermont-Ferrand, France; gleguenno@chu-clermontferrand.fr (G.L.G.); mruivard@chu-clermontferrand.fr (M.R.); 2Clermont Auvergne University, CNRS, SIGMA Clermont, Institute Pascal, Campus universitaire des Cézeaux, 4 Avenue Blaise Pascal, 63178 Aubière, France; laurent.sarry@uca.fr; 3LISIC Laboratory, Côte d’Opale University, 50 Rue Ferdinand Buisson, 62228 Calais, France; dehos@lisic.univ-littoral.fr; 4University Hospital Clermont-Ferrand, Biostatistics Unit, 58 Rue Montalembert, 63003 Clermont-Ferrand, France

**Keywords:** sclera, iron metabolism disorders, anemia, ROC curve, smartphone, diagnostic imaging

## Abstract

Iron deficiency (ID) is the most common nutritional deficiency. ID diagnosis requires ferritin measurement because clinical findings are poor and nonspecific. We studied the diagnostic value of blue sclera, which was scarcely reported as a specific and sensitive sign of ID. We enrolled 74 patients suspected of having ID. Pictures of their eyes were taken using a smartphone under similar daylight conditions. Three independent physicians graded the scleral color, and a computer analysis yielded the blue percentile of the sclera image. Final analysis included 67 patients (mean age 59.9 ± 20.1 years). Fifty-one had ID. Subjective blue scleral color was associated with ID for physician 1 (64.5% vs. 86.1%, *p* = 0.03). Sensitivity was 60.8% (CI95: 46.1%; 74.2%), specificity 68.8% (CI95: 41.3%; 89%), and positive predictive value 86.1% (CI95: 70.5%; 95.3%). A marginal difference was observed for other physicians (*p* = 0.05). Computer analysis showed higher blue in the ID group (*p* = 0.04). The area under the receiver operating characteristic (ROC) curve was 0.7 (0.54; 0.85). Sensitivity was 78.4% (CI95: 63.7%; 88.7%), specificity was 50% (CI95: 24.7%; 75.3%). Assessment of blue sclera was not influenced by iris color, sex, or anemia. We showed that blue sclera has good positive predictive value for ID diagnosis, and computer analysis was correlated to clinical assessment. Improvement of this innovative, non-invasive method could provide an easy handling and inexpensive diagnosis tool for ID.

## 1. Introduction

Iron deficiency (ID) is the world’s most common form of nutritional deficiency [1], with a prevalence of 4.5% to 18% [2]. ID remains the most common cause of anemia [3], with an annual incidence of 7.2–13.96 per 1000 person-years [4]. ID leads to disability through impairment of cognitive function [5], functional capacity, and quality of life [6,7], and it can be improved by iron supplementation. Consequently, ID is a major public health issue. 

ID diagnosis is based on the serum ferritin level, which is the most specific and sensitive test [8,9] with a cutoff of ≤30 or ≤100 µg/L in cases of ID accompanied by inflammation. Clinical findings in ID anemia are common to all types of anemia (fatigue, tachycardia [10]). For example, mucosal pallor was reported to have a sensitivity of 59% and a specificity of 63% for anemia diagnosis [11]. Iron is involved in multiple functions of epithelial cells, which are characterized by their high turnover. As such, various mucocutaneous manifestations are reported to be caused by ID, such as skin dryness, mild alopecia, atrophic glossitis, or koilonychia [12,13], but their specificity and sensitivity remain unknown. 

Among these signs, blueish coloration of the sclera was first described in 1908 by Osler in iron-deficient, malnourished teenagers and then sporadically reported [14]. In the late 1980s, Kalra et al. [15,16], Clemente et al. [17], and Kotsev et al. [18] published studies claiming that blue sclera was a highly sensitive and specific sign of ID anemia. In both studies, assessment of the blueish coloration of the sclera relied on subjective appreciation by physicians. Blue sclera can be associated with a wide variety of conditions [19] such as collagen diseases, for example in osteogenesis imperfecta, but these conditions are less common than ID. Recently, a clinical report of blue sclera has brought to the fore this clinical feature [20,21].

Despite these encouraging results, the use of blue sclera has only rarely been reported, and its assessment could be an inexpensive and effective diagnostic tool for ID. Therefore, we conducted a pilot study to determine the value of clinical blue assessment of blue sclera for the diagnosis of ID in a population of suspected ID. This study has two main objectives:(i)to study the relationship between physician assessment of blue sclera and iron stores;(ii)to provide an objective measurement of the blue color through computed analyses of eye images obtained using a smartphone.

## 2. Experimental Section

### 2.1. Ethics

We conducted a pragmatic, prospective, monocentric study in Clermont-Ferrand University Hospital. In line with French regulations, ethical approval was obtained from the International Review Board “CPP Sud EST VI” (ref. number 2019/CE02). The study was conducted in accordance with the Declaration of Helsinki and the Good Clinical Practice Recommendations.

### 2.2. Patients 

The inclusion criteria were inpatients aged ≥18 years, with:-Suspicion of ID as referral reason or discovered during the hospitalization regardless of the referral reason:
○clinical suspicion of anemia and/or ID, supported by pruritus, fatigue, pallor, tachycardia, restless leg syndrome, or glossitis;○biological suspicion ID anemia defined by microcytic hypochromic anemia or normocytic anemia with low reticulocyte count.-Biological assessment of serum iron stores by serum ferritin (SF) or transferrin percent saturation (TPS).

Exclusion criteria were medical history of Marfan or Ehler–Danlos syndrome, scurvy or osteogenesis imperfecta, chronic kidney disease (serum creatinine > 150 µmol/L or estimated glomerular filtration rate by CKD-EPI < 30 mL/min), or eye surgery or injury. 

As the entry point of the study was iron deficiency suspicion in patients referred to our internal medicine department, a control group of healthy volunteers would not have been appropriate.

### 2.3. Creation of the Smartphone Application and Blue Color Software Analyzer

First, a smartphone application was created using OpenCV library (Intel, Santa Clara, CA, USA) to provide repeatable image acquisition using multiple time exposure. We decided to use a smartphone device (Google Nexus 5X^®^, Mountain View, CA, USA) instead of a powerful professional camera because the aim was to allow widespread use of this diagnostic tool. The smartphone application allowed us to anonymize the pictures and to collect clinical and biological data for each patient. During a preliminary study conducted using the first nine patients enrolled [22], we tested the application in real conditions in our department, to determine an easy-handling and homogeneous image acquisition procedure. Briefly, several parameters were considered: creation of an automated sequence of multiple-time exposure image acquisition, deactivation of automated white color correction and white balance, and use of natural daylight with all artificial lights turned off. To measure the true color of a picture, we needed a white color reference. We first tried to use the reflection of natural daylight in the sclera, but this method was too aleatory. We finally decided to use a white reference provided by a color-checker (X-rite^®^, Grand Rapids, Michigan, USA), which was placed by the physician next to the eye, to take a picture including both the eye and white patch of the color-checker. 

Second, postprocessing software was created to analyze the blue color of the sclera: the computer code used a Python (Python software foundation, Wilmington, Delaware, USA) script fully available online at https://github.com/lasarry.com and was created by the department of one of the authors (L.S.). To obtain pictures with a highly dynamic range, images with multiple time exposures were merged into a single image. The software analyzed the percentage of each color (red, green, and blue) in scleral pixels after white correction. The software uses several data sets, including acquisition parameters of the images and coordinates of points inside the white color square and inside the sclerae, which are manually entered. Briefly, the script extracts the white color square and the sclera by region growing and gives output images. Then, the script takes the created output images as input, normalizes color inside the sclera with the white square value, and produces the 75% percentile of blue (BP) for each image. 

### 2.4. Data Collection

The pictures were taken by the same physician throughout the study, who collected, at the same time, clinical (age, sex, height, weight, body mass index; BMI) and biological data. The physician was not involved in the later scleral color assessment process. Exposure conditions were standardized: the patient sat next to a window in natural daylight. There was no direct contact with the patient, so the process was completely noninvasive. Iris color was graded as light (grey, green, or blue) or dark. 

Clinical assessment of blue sclera was conducted by three well-trained physicians of our department who were familiar with the study protocol. We decided to include only three physicians for three reasons: first, to be as close as possible to the process described in previous published studies [15,16,17,18]; second, because increasing the number of raters risks decreasing the statistical power because the inter-rater agreement in subjective assessment of clinical features is usually poor; third, because as a consequence of the previous point, it appears necessary that the physicians are familiar with scleral color evaluation. The physicians evaluated independently the color for each patient in anonymized pictures to ensure blindness from the patient’s conditions. The sequence of evaluation was randomized for each physician. Blue color was evaluated as 1 (absent), 2 (equivocal), 3 (definite), and 4 (striking).

### 2.5. Outcome Assessment

ID was defined as stated by Camaschella and colleagues [23] by one of the following criteria: -SF ≤ 30 µg/L, or ≤100 µg/L in the case of inflammatory syndrome,-TPS ≤ 16% and SF ≤ 300 µg/L.

Subjective blue color of the sclera was defined by a sign graded as definite or striking. We also used the definition used in previous studies, in that at least two physicians had to state that the sign as definite or striking [15,16,17,18]. Objective blue color was measured by computer analysis as the percentage of blue color in the 25% bluest pixel in the area of interest, which can be interpreted as a blue percentile (BP) in the analyzed pixels. 

We used three methods to delineate the area of interest of the sclera as follows:-Automated method—the sclera is automatically recognized by the software, and the whole sclera is analyzed. This method is termed full sclera (FS).-Semi-automated method—the physician selects the bluest area of the sclera to allow the software to delineate a small area of pixels with the same color characteristics. This method is termed semi-automated sclera (SAS).-Manual method—the analyzed area of sclera is manually delimited by the physician who delineates the bluest area of the sclera. This method is termed manual sclera (MS).
Examples of native and post-processed pictures are shown in Figure 1:

### 2.6. Statistical Analysis 

Sample size was determined to guarantee a satisfactory statistical power (i) to evaluate the reproducibility between physicians’ assessments of sclera color and (ii) to study the relationship between physician assessment of blue sclera and iron stores. Concerning the reproducibility, it is usually recommended to include a sample greater than 50 patients to highlight a concordance coefficient at least 0.70 [24]. Concerning the relationship between physician assessment of blue sclera and iron stores, 60 to 70 patients would allow to show an absolute difference of 25% between ID and no ID (NID), for a one-sided type I error at 5% and a statistical power at 80%. According to these considerations, it seemed that a sample size of 60 could be interesting in order to satisfactorily evaluate the primary objective of this study.

All statistical analyses were performed using Stata version 13 (StataCorp, College Station, TX, USA), for a type I error at 5%. The categorical parameters were expressed as numbers and percentages, whereas the quantitative variables were described as mean ± standard deviation or median (with interquartile range), according to the statistical distribution. Normality was studied using the Shapiro–Wilk test. The inter-rater reliability and agreement with the gold standard were evaluated using Lin’s concordance coefficient for quantitative variables and Fleiss’s κ coefficient for categorical data. The results were studied according to the usual rules defined in the literature [25]: 0.00–0.20 (poor agreement), 0.21–0.40 (fair agreement), 0.41–0.60 (moderate agreement), 0.61–0.80 (good agreement), and >0.80 (very good agreement). Then, the between-group comparisons (ID vs. NID) were performed using the following statistical tests: Student’s *t*-test, or Mann–Whitney test if the assumptions of the *t*-test were not met (normality, and homoscedasticity as identified by the Fisher–Snedecor test) for quantitative variables; and χ^2^ test or, if necessary, Fisher’s exact test for categorical variables. Statistical analyses were completed by analyzing the receiver operating characteristic (ROC) curve to determine the most appropriate threshold to predict ID for each method of FS, SAS, and MS. The optimal threshold was determined according to usual indexes reported in medical literature (Youden, Liu and efficiency). Then, sensitivity, specificity, and predictive values were estimated and expressed with 95% confidence intervals. Finally, to study the relationship between assessment of blue sclera and iron stores, a multivariable analysis, using logistic regression, was conducted to consider possible confounders determined according to clinical and biological relevance including age, gender, hemoglobin, and C-reactive protein (CRP). The results were expressed as an odds ratio (OR) and confidence interval 95% (CI95).

## 3. Results

From November 2014 to July 2018, 83 consecutive inpatients were enrolled. A flowchart is available in the Appendix A. As previously stated, we adjusted the technical settings on the camera to provide adequate picture quality using the first nine patients enrolled. These patients were not included in the final analysis because of the non-homogeneous shooting procedure: 6 of them had no pictures with white patches identified by the color-checker, and 3 of them did not have the multiple time-exposures required to create high dynamic range images. During the second phase, we tested 74 patients using the same procedure for shooting and the same camera settings. We excluded 7 patients: 4 because of poor picture quality (blurred images, over- or underexposed images), and 3 because of a lack of biological data regarding iron stores. A total of 67 were included in the final analysis. 

### 3.1. Inclusion Characteristics

The male:female ratio was 0.8 in the overall population and was significantly lower in the ID group (0.59) than in the NID group (2.2). Mean age was significantly higher in the ID group (*p* = 0.01). Sixty-three patients were Caucasian, three were North-African, and one was Asian. Overall, 52/67 (77.6%) patients had anemia (hemoglobin ≤12 g/dL for women, ≤13 g/dL for men): 46/51 in the ID group and 7/16 in the NID group. Baseline characteristics are shown in Appendix A.

### 3.2. Physician Assessment of Blue Sclera

We first compared physician assessments of sclera color and iron stores (ID or NID) (Table 1). We found a significant association between subjective blue color assessment of the sclera by physician 1 and the diagnosis of ID (*p* = 0.03). Multivariable logistic regression adjusted for age, gender, hemoglobin, and CRP confirmed this result (OR 2.39 (CI95: 1.08; 5.27), *p* = 0.03). However, we found a marginal difference for physicians 2 and 3 (*p* = 0.05).

Subjective analysis of blue color was not influenced for any of the physicians by iris color (*p* = 0.28, *p* = 0.37, and *p* = 0.44, respectively), sex (*p* = 0.58, *p* = 0.23, and *p* = 0.50, respectively), or the presence of anemia (*p* = 0.22, *p* = 0.08, and *p* = 0.07, respectively). 

Sensitivity was low: 60.8% (CI95: 46.1%; 74.2%), 45.1% (CI95: 31.1%; 59.7%), and 29.4% (CI95: 17.5%; 43.8%) for physicians 1, 2, and 3, respectively. Specificity was 68.8% (CI95: 41.3%; 89%), 81.3% (CI95: 54.4%; 96%), and 93.8% (CI95: 69.8%; 99.8%) for physicians 1, 2, and 3, respectively. Positive predictive value (PPV) was 86.1% (CI95: 70.5%; 95.3%), 88.5% (CI95: 69.8%; 97.6%), and 93.8% (CI95: 69.8%; 99.8%) for physicians 1, 2, and 3 respectively. Negative predictive value (NPV) was low: 35.5% (CI95: 19.2%; 54.6%), 31.7% (CI95: 18.1%; 48.1%), and 29.4% (CI95: 17.5%; 43.8%) for physicians 1, 2, and 3, respectively. Positive likelihood ratio (LR) was 1.95 (CI95: 0.91; 4.16), 2.41 (CI95: 0.83; 6.97), and 4.71 (CI95: 0.67; 32.9) for physicians 1, 2, and 3, respectively. 

When we used the definition of blue sclera found in previous studies, we found no significant association between blue sclera and ID (*p* = 0.09). This result could be linked to a poor inter-rater reliability, as shown by the Fleiss’s κ value, which was between 0.09 and 0.23. Sensitivity was 51% (CI95: 36.6%; 56.2%), specificity was 93.8% (CI95: 69.8%; 99.8%), PPV was 96.3% (CI95: 81%; 99.9%), and NPV 37.5% (CI95: 22.7%; 54.2%). Positive LR was 8.16 (CI95: 1.2; 55.5).

### 3.3. Computed Analysis of Blue Color

The concordance between the three methods of computed sclera color analysis was excellent, as shown by the Lin’s concordance correlation coefficient: *r* = 0.929 (FS vs. MS), *r* = 0.927 (FS vs. SAS), and *r* = 0.945 (MS vs. SAS). 

We found no effect of iris color on computed sclera color via any of the three methods (FS: *p* = 0.68; MS: *p* = 0.88; SAS: *p* = 0.94). As 94% of our patients were Caucasian, we did not perform any analysis according to skin color. Computed results of blue color analysis are shown in Table 2. 

Blue color seemed to be higher in ID than in NID subjects for all three methods, but the difference reached significance only for the MS method (*p* = 0.04) (FS method: *p* = 0.12; and SAS method: *p* = 0.05). Multivariable analyses confirmed these results (OR 1.28 (CI95: 1.03; 1.58), *p* = 0.02), with adjustment on age, gender, hemoglobin, and CRP levels

For physicians 1 and 3, the MS-computed blue color was significantly higher in patients with blue sclera (graded as definite or striking) compared to patients without blue sclera (*p* = 0.002 and *p* < 0.001 respectively, Table 3). For physician 2, the difference did not reach statistical significance (*p* = 0.33).

### 3.4. ROC Curves for Computed Blue Color

For the detection of ID, the ROC curve with the FS method (Figure 2a) showed a sensitivity of 90.2% (CI95: 78.6%; 96.7%), a specificity of 43.8% (CI95: 19.8%; 70.1%), PPV of 83.6% (CI95: 71.2%; 92.2%), and NPV of 58.3% (CI95: 27.7%; 84.8%) with a threshold of 27.6 (Youden index). 

The MS method (Figure 2b) showed a sensitivity of 78.4% (CI95: 64.7%; 88.7%), a specificity of 50% (CI95: 24.7%; 75.3%), PPV of 83.3% (CI95: 69.8%; 92.5%), and NPV of 42.1% (CI95: 20.3%; 66.5%) with a threshold of 29.1 and the highest area under the curve of 0.7 (CI95: 0.54; 0.85). 

The SAS method (Figure 2c) showed a sensitivity of 84.3% (CI95: 71.4%; 93%), a specificity of 50% (CI95: 24.7%; 75.3%), PPV of 84.3% (CI95: 71.4%; 93%), and NPV of 50% (CI95: 24.7%; 75.3%) with a threshold of 28.9. 

## 4. Discussion

To our best knowledge, ours is the first attempt to provide an objective measurement of blue sclera as a diagnostic tool for ID. We performed a two-step study. First, we evaluated subjective blue color assessment by three well-trained physicians who were blinded to the results of iron-store analysis. Our results are at odds with previously published work [14]. In our study, PPV was high (86.1%–93.8% depending on the physician), but sensitivity and specificity were low. One explanation could be the difference in inclusion criteria. We included patients based on clinical or biological suspicion of ID, while previous studies included patients with and without suspicion of anemia or ID. Furthermore, assessment of blue color was different in our study to ensure blindness of the physicians. Color assessment was undertaken on anonymized pictures in our study, while it was live in previous studies. Second, we applied a computer analysis of blue scleral color; blue color was higher in patients graded with blue sclera by physicians. Results were contradictory depending on the methodology used to delineate the sclera. The best results were obtained with the MS method, but specificity was poor with the three methods. These results might be due to interference color signals induced by the presence of blood vessels in the analyzed area of interest, enhancing the amount of red color. Although MS requires physician intervention, this method helps to avoid some image oddities that could reduce the quality of the computer analysis. 

Blue coloration of the sclera can be found in a wide variety of conditions such as antibiotic treatment by minocycline [26], alkaptonuria, or osteogenesis imperfecta (OI) [27]. In OI, quantitative or qualitative defects in type 1 collagen induces bone fragility. Notably, none of our patients had a history of multiple bone fractures or dysmorphic features. Electron microscopy of sclera tissues from OI patients revealed reduced collagen thickness of collagen fibers indicative of immature collagen, which is more translucent than normal collagen. This feature could explain the blueish coloration by making the blood vessels and underlying uveal pigment more visible [28,29]. Collagen structure of the sclera is composed of 90% type I collagen fibers [30], whose synthesis requires nonheme iron prolyl-4-hydroxylase [31]. We postulate that ID leads to impaired synthesis of type I collagen fibers reducing sclera thickness and resulting in blueish coloration. This hypothesis could be easily verified through non-invasive measurements of the sclera by anterior-segment optical coherent tomography [32]. Several limitations of our work can be raised. First, the study was conducted in a single center (tertiary medical care center). Second, only three physicians of the same department were involved in the clinical blue color assessment. Third, we did not evaluate the effect of iron supplementation on this clinical feature. Indeed, iron supplementation should restore normal collagen synthesis; the blue tint should disappear after iron correction, but, to date, there are no data in the literature to support this hypothesis. Further studies should focus on this question to improve the understanding of this feature. In line with this consideration, we suggest that including a control group of healthy volunteers would be also of interest to compare the diagnostic value of blue sclera for iron deficiency diagnosis in the general population.

In OI, blue sclera assessment was graded by comparison with a commercial scale of blue color. The grades ranged from 8 (bluest grade), which was a 50% dilution of the hue of a 4-year-old patient with OI, to 1 (almost white) by 50% dilution between each grade [33]. In our study, physicians graded the blue color in comparison with an absolute white color provided by a color-checker (X-Rite^®^), but the inter-rater reliability was poor. Any attempt to increase the complexity of the grading might reduce even further the extent of agreement between the assessors. Interestingly, iris color affected neither the assessment of the physician nor the computed analysis of the sclera color.

Moreover, the small size of our population meant that we could not perform an analysis according to ethnicity or to the presence of anemia, and we could not evaluate the impact of the severity of ID nor its duration. Finally, as a result of intrinsic efficiency characteristics of the test, computed study of blue color at the current state of development seems not efficient enough to provide wide use of this diagnosis method. However, the fair correlation between clinical assessment and computed evaluation is an encouraging result to develop this protocol further. 

## 5. Conclusions

Physician assessment of blue sclera has a good PPV for the diagnosis of ID in subjects with clinical or biological suspicion of ID. However, sensitivity and specificity are poor, and the inter-rater agreement is low. As such, in routine medical practice, the presence of blue sclera should encourage physicians to search for ID, but blood analysis remains the mainstay for ID diagnosis. 

Computed analysis provides encouraging results: blue color is higher in ID than in NID, but the diagnostic value depends on the methodology used to delineate the sclera. Further studies should focus on improving blue detection to enhance sensitivity and specificity. The improvement of this innovative method could provide an easy handling and inexpensive diagnosis tool of great interest for ID diagnosis.

## Figures and Tables

**Figure 1 jcm-08-01876-f001:**
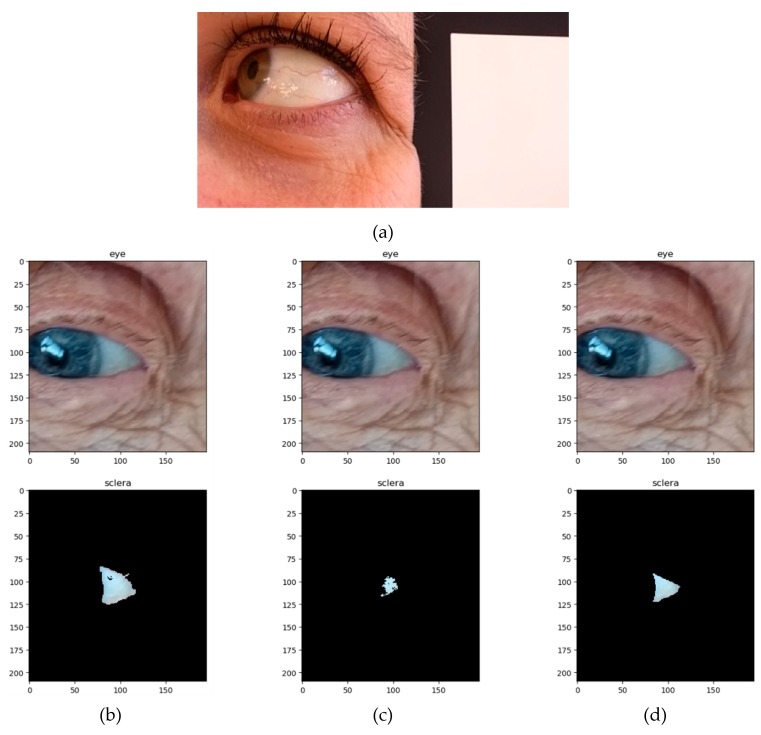
Examples of eye pictures taken during the study. (**a**) Native image of a 37-year-old woman without ID and no blue sclera with the white patch of the color checker. (**b–d**) Native and post-processed pictures of an 86-year-old woman with blue sclera and ID, according to the method for sclera delineation. (**b**) Automated method (full sclera, FS); (**c**) semi-automated method (semi-automated sclera, SAS); and (**d**) manual method (manual sclera, MS).

**Figure 2 jcm-08-01876-f002:**
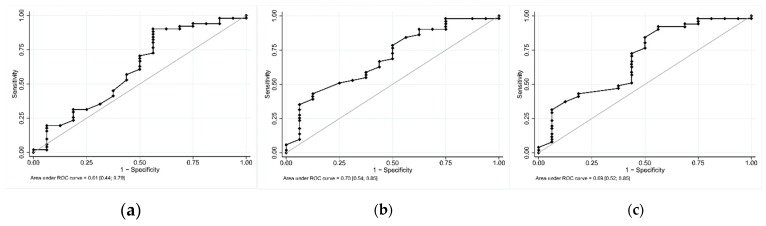
Receiver operating characteristic (ROC) curves of blue percentile for iron deficiency diagnosis. (**a**) Full sclera method; (**b**) manual sclera method; and (**c**) semi-automated sclera method.

**Table 1 jcm-08-01876-t001:** Physician assessment of blue sclera and iron stores.

		No Iron Deficiency (*n* = 16)	Iron Deficiency (*n* = 51)	*p* Value
**Physician 1**	Blue (%) (anemic)	**5 (13.89)** (3)	**31 (86.11)** (27)	*p* = 0.03
	No blue (%) (anemic)	**11 (35.48)** (4)	**20 (64.52)** (18)	
**Physician 2**	Blue (%) (anemic)	**3 (11.54)** (2)	**23 (88.46)** (17)	*p* = 0.05
	No blue (%) (anemic)	**13 (31.71)** (5)	**28 (68.29)** (24)	
**Physician 3**	Blue (%) (anemic)	**1 (6.25)** (1)	**15 (93.75)** (10)	*p* = 0.05
	No blue (%) (anemic)	**15 (29.41)** (6)	**36 (70.59)** (31)	

In bold, number and percent (in brackets) of patients graded as blue sclera and no blue sclera by physicians according to iron stores and anemia. NID: no iron deficiency; ID: iron deficiency.

**Table 2 jcm-08-01876-t002:** Computed blue color assessment according to the delineation method.

	NID (*n* = 16)	ID (*n* = 51)	*p* Value
**FS** All patients Anemic patients Nonanemic patients	**29.18 ± 4.31** 27.42 ± 4.78 30.55 ± 3.59	**31.07 ± 3.33** 30.77 ± 3.42 33.26 ± 1.12	*p* = 0.12
**MS** All patients Anemic patients Nonanemic patients	**29.12 ± 3.53** 27.74 ± 2.97 30.20 ± 3.71	**31.24 ± 3.39** 31.01 ± 3.51 33.00 ± 1.57	*p* =0.04
**SAS** All patients Anemic patients Nonanemic patients	**29.03 ± 3.87** 27.88 ± 4.02 29.93 ± 3.73	**31.19 ± 3.29** 30.99 ± 3.43 32.73 ± 1.16	*p* = 0.05

Results are presented as mean ± standard deviation of blue percentile (BP) according to iron stores for each method of scleral delineation (in bold, results for all patients regardless of the diagnosis of anemia). NID: no iron deficiency; ID: iron deficiency. FS: full sclera; MS: manual sclera; and SAS: semi-automated sclera.

**Table 3 jcm-08-01876-t003:** Computed blue sclera assessment by the MS method by physician.

	No Blue Sclera	Blue Sclera	*p* Value
**Physician 1** All patients Anemic patients Nonanemic patients	**29.27 ± 4.19** 28.85 ± 4.37 30.31 ± 3.76	**32.00 ± 2.17** 31.78 ± 2.22 33.32 ± 1.38	*p* = 0.002
**Physician 2** All patients Anemic patients Nonanemic patients	**30.42 ± 3.93** 30.24 ± 4.06 30.94 ± 3.67	**31.23 ± 2.75** 31.00 ± 2.81 33.00 ± 1.56	*p* = 0.33
**Physician 3** All patients Anemic patients Nonanemic patients	**30.01 ± 3.93** 29.56 ± 3.63 31.21 ± 3.37	**33.03 ± 1.93** 33.05 ± 2.00 NC (*n* = 1)	*p* < 0.001

Results are presented as mean ± standard deviation of blue percentile (BP) assessed by the manual sclera method according to physician assessment of sclera color (in bold, results for all patients regardless of the diagnosis of anemia). NC: not calculable. No significant difference in BP for any physician between anemic and nonanemic patients.

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
