# Peer review of "Computed and Subjective Blue Scleral Color Analysis as a Diagnostic Tool for Iron Deficiency: A Pilot Study"

_jcm, 2019, doi:10.3390/jcm8111876_

Round 1

Reviewer 1 Report

Dear author

Thank you for the opportunity to review this manuscipt. It was a pleasure.

The authors made several major revisions to significantly improve the quality of the manuscript, especially in the method section. 

Although the small size of the population study appears as one of the major limits,i strongly believe that the authors should proceed to the next
step study.

Thank you!

Best regards!

Author Response

Reviewer 1 comments:

Dear author

Thank you for the opportunity to review this manuscipt. It was a pleasure.

The authors made several major revisions to significantly improve the quality of the manuscript, especially in the method section. 

Although the small size of the population study appears as one of the major limits,i strongly believe that the authors should proceed to the next
step study.

Response to reviewer 1:

We want to thank the reviewer 1 for his kind comments and encouragements.

We are glad to hear that our revisions matched to the reviewer’s aspiration to increase the readability and the quality of the manuscript.

Dr Hervé Lobbes, on behalf of the coauthors.

Reviewer 2 Report

The introduction adequately presents the reader with an appropriate background, nevertheless the authors have not included other physical exam characteristics that can occur in iron-deficiency anemia as a comparison to bolster the proposition that this could be a useful addition. Like most deficiency-related conditions, there is a direct correlation in concentration of the missing metal or hormone (the greater the deficiency, the more likely one is to find an exam characteristic); however skin perfusion, capillary refill, overall skin pallor are known exam findings in iron deficiency anemia. When authors claim that other exam findings such as those listed above are "are poor and nonspecific," it would seem to me that their objective prior to publication would be to find an ocular sign that is not 'poor' or 'nonspecific.' It would benefit the reader to know what are the specificities and sensitivities of these skin findings relative to blue scleral hue. 

In their statistical analysis they do not present a sign that is specific (or sensitive for that matter), if one is to drill down to the numbers themselves. Furthermore, a 'blue' sclera can occur under many other circumstances. In fact, in their pictures they show patients who may also have age-related scleromalacia. They did not say that they exclude patients who may be taking medications such as amiodarone, phenothiazines, tetracycline, anti-malarial medications, oral or topical corticosteroids, epinephrine-containing or similar vasoactive medications (such as the very common phenylephrine). I would have liked to know if these patients were excluded. I would also like to know if patients had senile scleral plaques. Myopia is another common cause of a thin sclera and an unaccounted confounder which can make sclera appear blue due to the pathologic elongation of the axis of the eye and revelation of the underlying choroid. One can perform several tests in clinic to determine the thickness of the sclera including ultrasound and optical coherence tomography. Corneal pachymetry would also provide suggestive evidence of overall thinning indirectly in the sclera. Perhaps because this study was performed by internal medicine doctors, they would not be as familiar with these very common findings and confounders in an ophthalmology clinic.

The study was performed ethically, I see no conflicts of interest, and, importantly, I believe these kinds of uses of technology are of benefit to the medical public. Readers will be able to see for themselves in the statistical analysis that this exam finding has a low sensitivity and specificity relative to serum testing; and I applaud the authors for not over-selling their finding and pointing to their statistics in their conclusion.

Author Response

Response to reviewer 2 comments:

 Point 1: The introduction adequately presents the reader with an appropriate background, nevertheless the authors have not included other physical exam characteristics that can occur in iron-deficiency anemia as a comparison to bolster the proposition that this could be a useful addition. Like most deficiency-related conditions, there is a direct correlation in concentration of the missing metal or hormone (the greater the deficiency, the more likely one is to find an exam characteristic); however skin perfusion, capillary refill, overall skin pallor are known exam findings in iron deficiency anemia. When authors claim that other exam findings such as those listed above are "are poor and nonspecific," it would seem to me that their objective prior to publication would be to find an ocular sign that is not 'poor' or 'nonspecific.' It would benefit the reader to know what the specificities and sensitivities of these skin findings are relative to blue scleral hue. 

Response to point 1:

We thank reviewer 2 for his interesting suggestion. There are very few studies about the sensitivity and specificity of other signs of iron deficiency. Based on the study by Kalra et al [1], mucosal pallor, glossitis or koilonychia were less frequent than blue sclera in iron deficient patients (mucosal pallor 30%, glossitis 17%, koilonychia 4%, versus 87% for blue sclera). Moreover, these findings are common to all-type anemia regardless of the etiology. In a recent study by Kalantri et al [2], the accuracy of mucosal pallor for anemia diagnosis was poor: sensitivity ranged from 59% to 76% and specificity ranged from 63% to 64% according to the hemoglobin threshold (5 and 7 g/dL respectively).

Else ways, we did not include the evaluation of these clinical signs in our study to maintain the blindness of the physicians about iron stores: the scleral hue evaluation was performed on anonymized pictures, and evaluation of skin perfusion or skin pallor should have been performed by direct examination.

To help the reader to understand the potential interest of a better clinical sign, we added the following sentence as suggested:

Row 41-42… “For example, mucosal pallor was reported to have a sensitivity of 59% and a specificity of 63% for anemia diagnosis”

Row 45-46 “such as skin dryness, mild alopecia, atrophic glossitis, or koilonychia but their specificity and sensitivity remain unknown”.

Point 2: In their statistical analysis they do not present a sign that is specific (or sensitive for that matter), if one is to drill down to the numbers themselves. Furthermore, a 'blue' sclera can occur under many other circumstances. In fact, in their pictures they show patients who may also have age-related scleromalacia. They did not say that they exclude patients who may be taking medications such as amiodarone, phenothiazines, tetracycline, anti-malarial medications, oral or topical corticosteroids, epinephrine-containing or similar vasoactive medications (such as the very common phenylephrine). I would have liked to know if these patients were excluded. I would also like to know if patients had senile scleral plaques.

Response to point 2:

We thank reviewer 2 for his relevant comment. Statistical analysis revealed no influence of age on the blue percentile: this result is not in favor of blue color du to age-related scleromalacia. However, we agree that this data should be explored further during larger-scale study.

None of our patients were receiving tetracycline (such as minocycline which is known to induce blue coloration of the sclera), but we did not exclude the other medications cited above.

In the iron deficient population of our study, 17/51 (33%) patients had senile scleral plaques whereas in the non-iron deficient population, 3/16 (18.75%) had senile scleral plaques.

Point 3: Myopia is another common cause of a thin sclera and an unaccounted confounder which can make sclera appear blue due to the pathologic elongation of the axis of the eye and revelation of the underlying choroid. One can perform several tests in clinic to determine the thickness of the sclera including ultrasound and optical coherence tomography. Corneal pachymetry would also provide suggestive evidence of overall thinning indirectly in the sclera. Perhaps because this study was performed by internal medicine doctors, they would not be as familiar with these very common findings and confounders in an ophthalmology clinic.

Response to point 3:

We agree with the potential interest of measuring the sclera thickness to provide a “proof of concept” of our hypothesis. Although we pointed it in our discussion section (row 288), we are undoubtedly not very familiar with specific ophthalmological findings. Nevertheless, our aim was to provide a simple diagnosis tool requiring no specific material, to allow its widespread use.

Moreover, as pointed out by reviewer 2, as our study was conducted in internal medicine department, we were not able to perform this interesting study.

Point 4: The study was performed ethically, I see no conflicts of interest, and, importantly, I believe these kinds of uses of technology are of benefit to the medical public. Readers will be able to see for themselves in the statistical analysis that this exam finding has a low sensitivity and specificity relative to serum testing; and I applaud the authors for not over-selling their finding and pointing to their statistics in their conclusion.

Response to point 4:

We would like to warmly thank reviewer 2 for his kind comments and his encouragement.

Kalra, L.; Hamlyn, A.N.; Jones, B.J. Blue sclerae: a common sign of iron deficiency? Lancet 1986, 2, 1267–1269. Kalantri, A.; Karambelkar, M.; Joshi, R.; Kalantri, S.; Jajoo, U. Accuracy and Reliability of Pallor for Detecting Anaemia: A Hospital-Based Diagnostic Accuracy Study. PLoS One 2010, 5.